# Acceleration and Averaging
# In Stochastic Descent Dynamics

**Walid Krichene**
Google, Inc.
walidk@google.com

**Peter Bartlett**
U.C. Berkeley
bartlett@cs.berkeley.edu

## Abstract

We formulate and study a general family of (continuous-time) stochastic dynamics for accelerated first-order minimization of smooth convex functions.

Building on an averaging formulation of accelerated mirror descent, we propose a stochastic variant in which the gradient is contaminated by noise, and study the resulting stochastic differential equation. We prove a bound on the rate of change of an energy function associated with the problem, then use it to derive estimates of convergence rates of the function values (almost surely and in expectation), both for persistent and asymptotically vanishing noise. We discuss the interaction between the parameters of the dynamics (learning rate and averaging rates) and the covariation of the noise process. In particular, we show how the asymptotic rate of covariation affects the choice of parameters and, ultimately, the convergence rate.

## 1 Introduction

We consider the constrained convex minimization problem

$$\min_{x \in \mathcal{X}} f(x),$$

where $\mathcal{X}$ is a closed, convex, compact subset of $\mathbb{R}^n$, and $f$ is a proper closed convex function, assumed to be differentiable with Lipschitz gradient, and we denote $\mathcal{X}^\star \subset \mathcal{X}$ the set of its minimizers. First-order methods play an important role in minimizing such functions, in particular in large-scale machine learning applications, in which the dimensionality (number of features) and size (number of samples) in typical datasets makes higher-order methods intractable. Many such algorithms can be viewed as a discretization of continuous-time dynamics. The simplest example is gradient descent, which can be viewed as the discretization of the gradient flow dynamics $\dot{x}(t) = -\nabla f(x(t))$, where $\dot{x}(t)$ denotes the time derivative of a $C^1$ trajectory $x(t)$. An important generalization of gradient descent was developed by Nemirovsky and Yudin [1983], and termed mirror descent: it couples a dual variable $z(t)$ and its "mirror" primal variable $x(t)$. More specifically, the dynamics are given by

$$\text{MD} \begin{cases} \dot{z}(t) = -\nabla f(x(t)) \\ x(t) = \nabla \psi^*(z(t)), \end{cases} \quad (1)$$

where $\nabla \psi^* : \mathbb{R}^n \to \mathcal{X}$ is a Lipschitz function defined on the entire dual space $\mathbb{R}^n$, with values in the feasible set $\mathcal{X}$; it is often referred to as a mirror map, and we will recall its definition and properties in Section 2. Mirror descent can be viewed as a generalization of projected gradient descent, where the Euclidean projection is replaced by the mirror map $\nabla \psi^*$ [Beck and Teboulle, 2003]. This makes it possible to adapt the choice of the mirror map to the geometry of the problem, leading to better dependence on the dimension $n$, see [Ben-Tal and Nemirovski, 2001], [Ben-Tal et al., 2001].

**Continuous-time dynamics** Although optimization methods are inherently discrete, the continuous-time point of view can help in their design and analysis, since it can leverage the

rich literature on dynamical systems, control theory, and mechanics, see [Helmke and Moore, 1994], [Bloch, 1994], and the references therein. Continuous-time models are also commonly used in financial applications, such as option pricing [Black and Scholes, 1973], even though the actions are taken in discrete time. In convex optimization, beyond simplifying the analysis, continuous-time models have also motivated new algorithms: mirror descent is one such example, since it was originally motivated in continuous time (Chapter 3 in [Nemirovsky and Yudin, 1983]). In a more recent line of work ([Su et al., 2014], [Krichene et al., 2015], [Wibisono et al., 2016]), Nesterov's accelerated method [Nesterov, 1983] was shown to be the discretization of a second-order ordinary differential equation (ODE), which, in the unconstrained case, can be interpreted as a damped non-linear oscillator [Cabot et al., 2009, Attouch et al., 2015]. This motivated a restarting heuristic [O'Donoghue and Candès, 2015], which aims at further dissipating the energy. Krichene et al. [2015] generalized this ODE to mirror descent, and gave an averaging interpretation of accelerated dynamics by writing it as two coupled first-order ODEs. This is the starting point of this paper, in which we introduce and study a stochastic variant of accelerated mirror descent.

**Stochastic dynamics and related work** The dynamics that we have discussed so far are *deterministic* first-order dynamics, since they use the exact gradient $\nabla f$. However, in many machine learning applications, evaluating the exact gradient $\nabla f$ can be prohibitively expensive, e.g. in regularized empirical risk minimization problems, where the objective function $f$ involves the sum of loss functions over a training set, of the form $f(x) = \frac{1}{|\mathcal{I}|} \sum_{i \in \mathcal{I}} f_i(x) + g(x)$, where $\mathcal{I}$ indexes the training samples, and $g$ is a regularization function[1]. Instead of computing the exact gradient $\nabla f(x) = \frac{1}{|\mathcal{I}|} \sum_{i \in \mathcal{I}} \nabla f_i(x) + \nabla g(x)$, a common approach is to compute an unbiased, stochastic estimate of the gradient, given by $\frac{1}{|\tilde{\mathcal{I}}|} \sum_{i \in \tilde{\mathcal{I}}} \nabla f_i(x) + \nabla g(x)$, where $\tilde{I}$ is a uniformly random subset of $\mathcal{I}$, indexing a random batch of samples from the training set. This approach motivates the study of stochastic dynamics for convex optimization. But despite an extensive literature on stochastic gradient and mirror descent in discrete time, e.g. [Nemirovski et al., 2009], [Duchi et al., 2010], [Lan, 2012], [Johnson and Zhang, 2013], [Xiao and Zhang, 2014], and many others, few results are known for stochastic mirror descent in continuous-time. To the best of our knowledge, the only published results are by Raginsky and Bouvrie [2012] and Mertikopoulos and Staudigl [2016]. In its simplest form, the stochastic gradient dynamics can be described by the (underdamped) Langevin equation

$$dX(t) = -\nabla f(X(t)) + \sigma dB(t),$$

where $B(t)$ denotes a standard Wiener process (Brownian motion). It has a long history in optimization [Chiang et al., 1987], dating back to simulated annealing, and it is known to have a unique invariant measure with density proportional to the Gibbs distribution $e^{-\frac{2f(x)}{\sigma}}$ (see, e.g., [Pavliotis, 2014]). Langevin dynamics have recently played an important role in the analysis of sampling methods [Dalalyan, 2017, Bubeck et al., 2015, Durmus and Moulines, 2016, Cheng and Bartlett, 2017, Eberle et al., 2017, Cheng et al., 2017], where $f$ is taken to be proportional to the logarithm of a target density. It has also been used to derive convergence rates for smooth, non-convex optimization where the objective is dissipative [Raginsky et al., 2017].

For mirror descent dynamics, Raginsky and Bouvrie [2012] were the first to propose a stochastic variant of the mirror descent ODE (1), given by the SDE:

$$\text{SMD} \begin{cases} dZ(t) = -\nabla f(X(t)) + \sigma dB(t) \\ X(t) = \nabla \psi^*(Z(t)), \end{cases} \tag{2}$$

where $\sigma$ is a constant volatility. They argued that the function values $f(X(t))$ along sample trajectories do not converge to the minimum value of $f$ due to the persistent noise, but the optimality gap is bounded by a quantity proportional to $\sigma^2$. They also proposed a method to reduce the variance by simultaneously sampling multiple trajectories and linearly coupling them. Mertikopoulos and Staudigl [2016] extended the analysis in some important directions: they replaced the constant $\sigma$ with a general volatility matrix $\sigma(x,t)$ which can be space and time dependent, and studied two regimes: the small noise limit ($\sigma(x,t)$ vanishes at a $\mathcal{O}(1/\sqrt{\log t})$ rate), in which case they prove almost sure convergence; and the persistent noise regime ($\sigma(x,t)$ is uniformly bounded), in which case they define

a rectified variant of SMD, obtained by replacing the second equation by $X(t) = \nabla\psi^*(Z(t)/s(t))$, where $1/s(t)$ is a sensitivity parameter (intuitively, decreasing the sensitivity reduces the impact of accumulated noise). In particular, they prove that with $s(t) = \sqrt{t}$, the expected function values converge at a $\mathcal{O}(1/\sqrt{t})$ rate. While these recent results paint a broad picture of mirror descent dynamics, they leave many questions open: in particular, they do not provide estimates for convergence rates in the vanishing noise limit, which is an important regime in machine learning applications, since one can often control the variance of the gradient estimate, for example by gradually increasing the batch size, as done by Xiao and Zhang [2014]. Besides, they do not study accelerated dynamics, and the interaction between acceleration and noise remains unexplored in continuous time.

**Our contributions**   In this paper, we answer many of the questions left open in previous works. We formulate and study a family of stochastic accelerated mirror descent dynamics, and we characterize the interaction between its different parameters: the volatility of the noise, the (primal and dual) learning rates, and the sensitivity of the mirror map. More specifically:

- In Theorem 1, we give sufficient conditions for almost sure convergence of solution trajectories to the set of minimizers $\mathcal{X}^\star$. In particular, we show that it is possible to guarantee almost sure convergence even when the volatility grows unbounded asymptotically.
- In Theorem 2, we derive a bound on the expected function values. In particular, we can prove that in the vanishing noise regime, acceleration (with appropriate averaging) achieves a faster rate, see Corollary 2 and the discussion in Remark 3.
- In Theorem 3, we provide estimates of sample trajectory convergence rates.

The rest of the paper is organized as follows: We review the building blocks of our construction in Section 2, then formulate the stochastic dynamics in Section 3, and prove two instrumental lemmas. Section 4 is dedicated to the convergence results. We conclude with a brief discussion in Section 5.

## 2   Accelerated Mirror Descent Dynamics

### 2.1   Smooth mirror map

We start by reviewing some definitions and preliminaries. Let $(E, \|\cdot\|)$, be a normed vector space, and $(E^*, \|\cdot\|_*)$ be its dual space equipped with the dual norm, and denote by $\langle x, z \rangle$ the pairing between $x \in E, z \in E^*$. To simplify, both $E$ and $E^*$ can be identified with $\mathbb{R}^n$, but we make the distinction for clarity. We say that a map $F : E \to E^*$ is Lipschitz continuous on $\mathcal{X} \subset E$ with constant $L$ if for all $x, x' \in \mathcal{X}$, $\|F(x) - F(x')\|_* \leq L\|x - x'\|$. Let $\psi : E \to \mathbb{R} \cup \{+\infty\}$ be a convex function with effective domain $\mathcal{X}$ (i.e. $\mathcal{X} = \{x \in E : \psi(x) < \infty\}$). Its convex conjugate $\psi^*$ is defined on $E^*$ by $\psi^*(z) = \sup_{x \in \mathcal{X}} \langle z, x \rangle - \psi(x)$. One can show that if $\psi$ is strongly convex, then $\psi^*$ is differentiable on all of $E^*$, and its gradient $\nabla\psi^*$ is a Lipschitz function that maps $E^*$ to $\mathcal{X}$ (see the supplementary material). This map is often called a mirror map [Nemirovsky and Yudin, 1983]. To give a concrete example, take $\psi$ to be the squared Euclidean norm, $\psi(x) = \frac{1}{2}\|x\|_2^2$. Then one can show $\psi^*(z) = \arg\min_{x \in \mathcal{X}} \|z - x\|_2^2$, and the mirror map reduces to the Euclidean projection on $\mathcal{X}$. For additional examples, see e.g. Banerjee et al. [2005]. We make the following assumptions throughout the paper:

**Assumption 1.** *$\mathcal{X}$ is closed, convex and compact, the set of minimizers $\mathcal{X}^\star$ is contained in the relative interior of $\mathcal{X}$, $\psi$ is non-negative (without loss of generality), $\psi^*$ is twice differentiable with a Lipschitz gradient, and $f$ is differentiable with a Lipschitz gradient. We denote by $L_{\psi^*}$ the Lipschitz constant of $\nabla\psi^*$, and by $L_f$ the Lipschitz constant of $\nabla f$.*

### 2.2   Averaging formulation of accelerated mirror descent

We start from the averaging formulation of Krichene et al. [2015], and include a sensitivity parameter similar to Mertikopoulos and Staudigl [2016]. This results in the following ODE:

$$\text{AMD}_{\eta,a,s} \begin{cases} \dot{z}(t) = -\eta(t)\nabla f(x(t)) \\ \dot{x}(t) = a(t)(\nabla\psi^*(z(t)/s(t)) - x(t)), \end{cases} \tag{3}$$

with initial conditions[2] $(x(t_0), z(t_0)) = (x_0, z_0)$. The ODE system is parameterized by the following functions, all assumed to be positive and continuous on $[t_0, \infty)$ (see Figure 1 for an illustration):

- $s(t)$ is a non-decreasing, inverse sensitivity parameter. As we will see, $s(t)$ will be helpful in the stochastic case in scaling the noise term, in order to reduce its impact.
- $\eta(t)$ is a learning rate in the dual space.
- $a(t)$ is an averaging rate in the primal space. Indeed, the second ODE in (3) can be written in integral form as a weighted average of the mirror trajectory as follows: let $w(t) = e^{\int_{t_0}^{t} a(\tau)d\tau}$ (equivalently, $a(t) = \frac{\dot{w}(t)}{w(t)}$), then the ODE is equivalent to $w(t)\dot{x}(t) + \dot{w}(t)x(t) = \dot{w}(t)\nabla\psi^*(z(t)/s(t))$, and integrating and rearranging,

$$x(t) = \frac{x(t_0)w(t_0) + \int_{t_0}^{t} \dot{w}(\tau)\nabla\psi^*(Z(\tau)/s(\tau))d\tau}{w(t)}.$$

There are other, different ways of formulating the accelerated dynamics: instead of two first-order ODEs, one can write one second-order ODE (such as in Su et al. [2014], Wibisono et al. [2016]), which has interesting interpretations related to Lagrangian dynamics. The averaging formulation given in Equation (3) is better suited to our analysis.

## 2.3 Energy function

The analysis of continuous-time dynamics often relies on a Lyapunov argument (in reference to Lyapunov [1892]): one starts by defining a non-negative energy function, then bounding its rate of change along solution trajectories. This bound can then be used to prove convergence to the set of minimizers $\mathcal{X}^\star$. We will consider a modified version of the energy function used in Krichene et al. [2016]: given a positive, $C^1$ function $r(t)$, and a pair of optimal primal-dual points $(x^\star, z^\star)$ such that[3] $x^\star \in \mathcal{X}^\star$ and $\nabla\psi^*(z^\star) = x^\star$, let

$$L(x, z, t) = r(t)(f(x) - f(x^\star)) + s(t)D_{\psi^*}(z(t)/s(t), z^\star). \tag{4}$$

Here, $D_{\psi^*}$ is the Bregman divergence associated with $\psi^*$, defined by

$$D_{\psi^*}(z', z) = \psi^*(z') - \psi^*(z) - \langle\nabla\psi^*(z), z' - z\rangle, \qquad \text{for all } z, z' \in E^*.$$

Then we can prove a bound on the time derivative of $L$ along solution trajectories of $\text{AMD}_{\eta,a,s}$, given in the following proposition. To keep the equations compact, we will occasionally omit explicit dependence on time, and write, e.g. $\eta/r$ instead of $\eta(t)/r(t)$.

**Lemma 1.** *Suppose that $a = \eta/r$. Then under $\text{AMD}_{\eta,\eta/r,s}$, for all $t \geq t_0$,*

$$\frac{d}{dt}L(x(t), z(t), t) \leq (f(x(t)) - f(x^\star))(\dot{r}(t) - \eta(t)) + \psi(x^\star)\dot{s}(t). \tag{5}$$

*Proof.* We start by bounding the rate of change of the Bregman divergence term:

$$\frac{d}{dt}s(t)D_{\psi^*}(z(t)/s(t), z^\star) = \dot{s}D_{\psi^*}(z/s, z^\star) + s\langle\nabla\psi^*(z/s) - \nabla\psi^*(z^\star), \dot{z}/s - \dot{s}z/s^2\rangle$$

$$= \langle\nabla\psi^*(z/s) - x^\star, \dot{z}\rangle + \dot{s}(D_{\psi^*}(z/s, z^\star) - \langle\nabla\psi^*(z/s) - \nabla\psi^*(z^\star), z/s\rangle)$$

$$= \langle\nabla\psi^*(z/s) - x^\star, \dot{z}\rangle + \dot{s}(\psi(x^\star) - \psi(\nabla\psi^*(z/s)))$$

$$\leq \langle\nabla\psi^*(z/s) - x^\star, \dot{z}\rangle + \dot{s}\psi(x^\star),$$

where the third equality can be proved using the fact that $\psi(x) + \psi^*(z) = \langle x, z\rangle \Leftrightarrow x \in \partial\psi^*(z) \Leftrightarrow z \in \partial\psi(x)$ (Theorem 23.5 in Rockafellar [1970]), and the last inequality follows from the assumption that $s$ is non-decreasing, and that $\psi$ is non-negative. Using this expression, we can then compute

$$\frac{d}{dt}L(x(t), z(t), t) \leq \dot{r}(f(x) - f(x^\star)) + r\langle\nabla f(x), \dot{x}\rangle + \langle\nabla\psi^*(z/s) - x^\star, \dot{z}\rangle + \psi(x^\star)\dot{s}$$

$$= \dot{r}(f(x) - f(x^\star)) + r\langle\nabla f(x), \dot{x}\rangle + \langle\dot{x}/a + x - x^\star, -\eta\nabla f(x)\rangle + \psi(x^\star)\dot{s}$$

$$\leq (f(x) - f(x^\star))(\dot{r} - \eta) + \langle\nabla f(x), \dot{x}\rangle(r - \eta/a) + \psi(x^\star)\dot{s},$$

where we plugged in the expression of $\dot{z}$ and $\nabla\psi^*(z/s)$ from $\mathrm{AMD}_{\eta,a,s}$ in the second equality, and used convexity of $f$ in the last inequality. The assumption $a = \eta/r$ ensures that the middle term vanishes, which concludes the proof. $\qquad\square$

As a consequence of the previous proposition, we can prove the following convergence rate:

**Corollary 1.** *Suppose that $a = \eta/r$ and that $\eta \geq \dot{r}$. Then under $\mathrm{AMD}_{\eta,\eta/r,s}$, for all $t \geq t_0$*

$$f(x(t)) - f(x^\star) \leq \frac{\psi(x^\star)(s(t) - s(t_0)) + L(x_0, z_0, t_0)}{r(t)}.$$

*Proof.* Starting from the bound (5), the first term is non-positive by the assumption that $\eta \geq \dot{r}$. Integrating, we have $L(x(t), z(t), t) - L(x_0, z_0, t_0) \leq \psi(x^\star)(s(t) - s(t_0))$, thus,

$$f(x(t) - f(x^\star)) \leq \frac{L(x(t), z(t), t)}{r(t)} \leq \frac{\psi(x^\star)(s(t) - s(t_0)) + L(x_0, z_0, t_0)}{r(t)}. \qquad\square$$

**Remark 1.** *Corollary 1 can be interpreted as follows: given a desired convergence rate $r(t)$, one can choose parameters $a, \eta, s$ that satisfy the conditions of the corollary (e.g. by first setting $\eta = \dot{r}$, then choosing $a = \eta/r$). This defines an ODE, the solutions of which are guaranteed to converge at the rate $r(t)$. While the convergence rate can seemingly be arbitrary for continuous time dynamics, discretizing the ODE does not always preserve the convergence rate. Wibisono et al. [2016], Wilson et al. [2016] give sufficient conditions on the discretization scheme to preserve polynomial rates, for example, a first-order discretization can preserve quadratic rates, and a higher-order discretization (using cubic-regularized Newton updates) can preserve cubic rates.*

**Remark 2.** *As a special case, one can recover Nesterov's ODE by taking $r(t) = t^2$, $\eta(t) = \beta t$, $a(t) = \beta/t$ (i.e. $w(t) = w(t_0)(t/t_0)^\beta$), and $s(t) = 1$ (see the supplement for additional details). It is worth observing that in this case, both the primal and dual rates $\eta(t)$ and $w(t)$ are increasing. A different choice of parameters leads to dynamics similar to Nesterov's but with different weights.*

## 3 Stochastic dynamics

We now formulate the stochastic variant of accelerated mirror descent dynamics (SAMD). Intuitively, we would like to replace the gradient term $\nabla f(x)$ in $\mathrm{AMD}_{\eta,a,s}$ by a noisy gradient. Writing the noisy dynamics as an Itô SDE [Øksendal, 2003], we consider the system

$$\mathrm{SAMD}_{\eta,a,s} \begin{cases} dZ(t) = -\eta(t)[\nabla f(X(t))dt + \sigma(X(t), t)dB(t)] \\ dX(t) = a(t)[\nabla\psi^*(Z(t)/s(t)) - X(t)]dt, \end{cases} \qquad (6)$$

with initial condition $(X(t_0), Z(t_0)) = (x_0, z_0)$ (we assume deterministic initial conditions for simplicity). Here, $B(t) \in \mathbb{R}^n$ is a standard Wiener process with respect to a given filtered probability space $(\Omega, \mathcal{F}, \{\mathcal{F}_t\}_{t \geq t_0}, \mathbb{P})$, and $\sigma : (x, t) \mapsto \sigma(x, t) \in \mathbb{R}^{n \times n}$ is a volatility matrix assumed measurable and Lipschitz in $x$ (uniformly in $t$), and continuous in $t$ for all $x$. The drift term in $\mathrm{SAMD}_{\eta,a,s}$ is identical to the deterministic case, and the volatility term $-\eta(t)\sigma(X(t), t)dB(t)$ represents the noise in the gradient. In particular, we note that the learning rate $\eta(t)$ multiplies $\sigma(X(t), t)dB(t)$, to capture the fact that the gradient noise is scaled by the learning rate $\eta$. This formulation is fairly general, and does not assume, in particular, that the different components of the noise are independent, as we can see in the quadratic covariation of the dual process $Z(t)$:

$$d[Z_i(t), Z_j(t)] = \eta(t)^2(\sigma(X(t), t)\sigma(X(t), t)^T)_{i,j}dt = \eta(t)^2\Sigma_{ij}(X(t), t)dt, \qquad (7)$$

where we defined the infinitesimal covariance matrix $\Sigma(x, t) = \sigma(x, t)\sigma(x, t)^T \in \mathbb{R}^{n \times n}$. In our analysis, we will focus on different noise regimes, which can be characterized using[4]

$$\sigma_*^2(t) = \sup_{x \in \mathcal{X}} \|\Sigma(x, t)\|_i, \qquad (8)$$

where $\|\Sigma\|_i = \sup_{\|z\|_* \leq 1} \|\Sigma z\|$ is the induced matrix norm. Since $\Sigma(x, t)$ is Lipschitz in $x$ and continuous in $t$, and $\mathcal{X}$ is compact, $\sigma_*(t)$ is finite for all $t$, and continuous. Contrary to [Raginsky and Bouvrie, 2012, Mertikopoulos and Staudigl, 2016], we do not assume that $\sigma_*(t)$ is uniformly bounded in $t$. We give an illustration of the stochastic dynamics in Figure 1 (see the supplement for details).

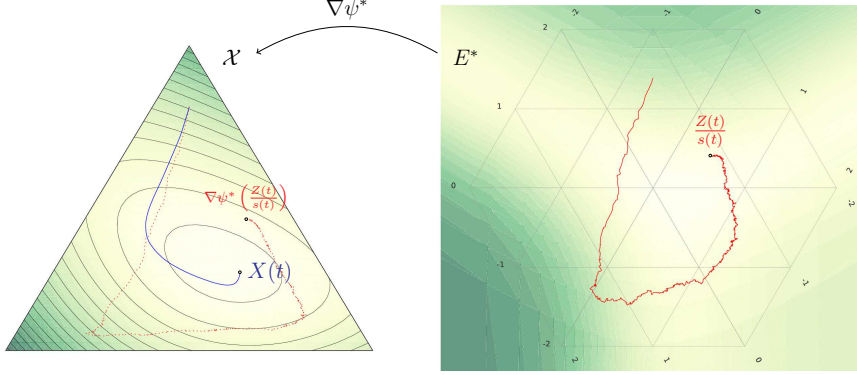

Figure 1: Illustration of the SAMD dynamics. The dual variable $Z(t)$ cumulates gradients. It is scaled by the sensitivity $1/s(t)$ then mapped to the primal space via the mirror map, resulting in $\nabla\psi^*(Z/s)$ (dotted line). The primal variable is then a weighted average of the mirror trajectory.

**Existence and uniqueness** First, we give the following existence and uniqueness result:

**Proposition 1.** *For all $T > t_0$, $\mathrm{SAMD}_{\eta,a,s}$ has a unique (up to redefinition on a $\mathbb{P}$-null set) solution $(X(t), Z(t))$ continuous on $[0, T]$, with the property that $(X(t), Z(t))$ is adapted to the filtration $\{\mathcal{F}_t\}$, and $\int_{t_0}^{T} \|X(t)\|^2 dt$, $\int_{t_0}^{T} \|Z(t)\|_*^2 dt$ have finite expectations.*

*Proof.* By assumption, $\nabla\psi^*$ and $\nabla f$ are Lipschitz continuous, thus the function $(x, z) \mapsto (-\eta(t)\nabla f(x), a(t)[\nabla\psi^*(z/s(t)) - x])$ is Lipschitz on $[t_0, T]$ (since $a, \eta, s$ are positive continuous). Additionally, the function $x \mapsto \sigma(x, t)$ is also Lipschitz. Therefore, we can invoke the existence and uniqueness theroem for stochastic differential equations [Øksendal, 2003, Theorem 5.2.1]. $\square$

Since $T$ is arbitrary, we can conclude that there exists a unique continuous solution on $[t_0, \infty)$.

**Energy decay** Next, in order to analyze the convergence properties of the solution trajectories $(X(t), Z(t))$, we will need to bound the time-derivative of the energy function $L$.

**Lemma 2.** *Suppose that the primal rate $a = \eta/r$, and let $(X(t), Z(t))$ be the unique solution to $\mathrm{SAMD}_{\eta,\eta/r,s}$. Then for all $t \geq t_0$,*

$$dL(X(t), Z(t), t) \leq \left[(f(X(t)) - f(x^\star))(\dot{r}(t) - \eta(t)) + \psi(x^\star)\dot{s}(t) + \frac{nL_{\psi^*}}{2}\frac{\eta^2(t)\sigma_*^2(t)}{s(t)}\right] dt + \langle V(t), dB(t)\rangle,$$

*where $V(t)$ is the continuous process given by*

$$V(t) = -\eta(t)\sigma(X(t), t)^T (\nabla\psi^*(Z(t)/s(t)) - \nabla\psi^*(z^\star)). \tag{9}$$

*Proof.* By definition of the energy function $L$, $\nabla_x L(x, z, t) = r(t)\nabla f(x)$ and $\nabla_z L(x, z, t) = \nabla\psi^*(z/s(t)) - \nabla\psi^*(z^\star)$, which are Lipschitz continuous in $(x, z)$ (uniformly in $t$ on any bounded interval, since $s, r$ are continuous positive functions of $t$). Thus by the Itô formula for functions with Lipschitz continuous gradients [Errami et al., 2002], we have

$$dL = \partial_t L dt + \langle\nabla_x L, dX\rangle + \langle\nabla_z L, dZ\rangle + \frac{1}{2}\mathrm{tr}\left(\eta\sigma^T\nabla_{zz}^2 L\sigma\eta\right) dt$$

$$= \partial_t L dt + \langle\nabla_x L, dX\rangle + \langle\nabla_z L, -\eta\nabla f(X)\rangle dt + \langle\nabla_z L, -\eta\sigma dB\rangle + \frac{\eta^2}{2}\mathrm{tr}\left(\Sigma\nabla_{zz}^2 L\right) dt.$$

The first three terms correspond exactly to the deterministic case, and we can bound them by (5) from Lemma 1. The last two terms are due to the stochastic noise, and consist of a volatility term

$$-\eta\langle\nabla_z L(X, Z, t), \sigma dB\rangle = -\eta\langle\nabla\psi^*(Z/s) - \nabla\psi^*(z^\star), \sigma dB\rangle = \langle V, dB\rangle,$$

and the Itô correction term

$$\frac{\eta^2}{2}\mathrm{tr}\left(\Sigma(X, t)\nabla_{zz}^2 L(X, Z, t)\right) dt = \frac{\eta^2}{2s}\mathrm{tr}\left(\Sigma(X, t)\nabla^2\psi^*(Z/s)\right) dt.$$

We can bound the last term using the fact that $\nabla\psi^*$ is, by assumption, $L_{\psi^*}$-Lipschitz, and the definition (8) of $\sigma^*$: for all $x \in E$, $z \in E^*$, and $t \geq t_0$, $\operatorname{tr}(\Sigma(x,t)\nabla^2\psi^*(z)) \leq nL_{\psi^*}\sigma_*^2(t)$. Combining the previous inequalities, we obtain the desired bound. $\qquad\square$

Integrating the bound of Lemma 2 will allow us to bound changes in energy. This bound will involve the Itô martingale term $\int_{t_0}^t \langle V(\tau), dB(\tau)\rangle$, and in order to control this term, we give, in the following lemma, an asymptotic envelope (a consequence of the law of the iterated logarithm).

**Lemma 3.** *Let* $b(t) = \int_{t_0}^t \eta^2(\tau)\sigma_*^2(\tau)d\tau$. *Then*

$$\int_{t_0}^t \langle V(\tau), dB(\tau)\rangle = \mathcal{O}(\sqrt{b(t)\log\log b(t)}) \qquad\qquad \text{a.s. as } t \to \infty. \qquad (10)$$

*Proof.* Let us denote the Itô martingale by $\mathcal{V}(t) = \int_{t_0}^t \langle V(\tau), dB(\tau)\rangle = \sum_{i=1}^n \int_{t_0}^t V_i(\tau)dB_i(\tau)$, and its quadratic variation by $\beta(t) = [\mathcal{V}(t), \mathcal{V}(t)]$. By definition of $\mathcal{V}$, we have

$$d\beta = \sum_{i=1}^n \sum_{j=1}^n V_i V_j d[B_i, B_j] = \sum_{i=1}^n V_i^2 dt = \langle V, V\rangle\, dt.$$

By the Dambis-Dubins-Schwartz time change theorem (e.g. Corollary 8.5.4 in [Øksendal, 2003]), there exists a Wiener process $\hat{B}$ such that

$$\mathcal{V}(t) = \hat{B}(\beta(t)). \qquad (11)$$

We now proceed to bound $\beta(t)$. Using the expression (9) of $V$, we have $\langle V, V\rangle = \eta^2(t)\Delta^T(t)\Sigma(X,t)\Delta(t)$, where $\Delta(t) = \nabla\psi^*(Z(t)/s(t)) - \nabla\psi^*(z^*)$. Since the mirror map has values in $\mathcal{X}$ and $\mathcal{X}$ is assumed compact, the diameter $D = \sup_{x,x'\in\mathcal{X}} \|x - x'\|$ is finite, and $\Delta(t) \leq D$ for all $t$. Thus, $d\beta(t) \leq D^2\eta(t)^2\sigma_*^2(t)dt$, and integrating,

$$\beta(t) \leq D^2 b(t) \text{ a.s.} \qquad (12)$$

Since $\beta(t)$ is a non-decreasing process, two cases are possible: if $\lim_{t\to\infty}\beta(t)$ is finite, then $\limsup_{t\to\infty}|\mathcal{V}(t)|$ is a.s. finite and the result follows immediately. If $\lim_{t\to\infty}\beta(t) = \infty$, then

$$\limsup_{t\to\infty} \frac{\mathcal{V}(t)}{\sqrt{b(t)\log\log b(t)}} \leq \limsup_{t\to\infty} \frac{\hat{B}(\beta(t))}{\sqrt{\frac{\beta(t)}{D^2}\log\log\frac{\beta(t)}{D^2}}} = D\sqrt{2} \qquad \text{a.s.}$$

where the inequality combines (11) and (12), and the equality is by the law of the iterated logarithm. $\qquad\square$

## 4 Convergence results

Equipped with Lemma 2 and Lemma 3, which bound, respectively, the rate of change of the energy and the asymptotic growth of the martingale term, we are now ready to prove our convergence results.

**Theorem 1.** *Suppose that* $\eta(t)\sigma_*(t) = o(1/\sqrt{\log t})$, *and that* $\int_{t_0}^t \eta(\tau)d\tau$ *dominates* $b(t)$ *and* $\sqrt{b(t)\log\log b(t)}$ *(where* $b(t) = \int_{t_0}^t \eta^2(\tau)\sigma_*^2(\tau)d\tau$ *as defined in Lemma 3). Consider* SAMD *dynamics with* $r = s = 1$. *Let* $(X(t), Z(t))$ *be the unique continuous solution of* $\text{SAMD}_{\eta,\eta,1}$. *Then*

$$\lim_{t\to\infty} f(X(t)) - f(x^*) = 0 \qquad\qquad \text{a.s.}$$

*Proof sketch.* We give a sketch of the proof here (the full argument is deferred to the supplement).

i) The first step is to prove that under the conditions of the theorem, the continuous solution of $\text{SAMD}_{\eta,\eta,1}$, $(X(t), Z(t))$, is an asymptotic pseudo trajectory (a notion defined and studied by Benaïm and Hirsch [1996] and Benaïm [1999]) of the deterministic flow $\text{AMD}_{\eta,\eta,1}$. The rigorous definition is given in the supplementary material, but intuitively, this means that for large enough times, the sample paths of the process $(X(t), Z(t))$ get arbitrarily close to $(x(t), z(t))$, the solution trajectories of the deterministic dynamics.

ii) The second step is to show that under the deterministic flow, the energy $L$ decreases enough for large enough times.

iii) The third step is to prove that under the stochastic process, $f(X(t))$ cannot stay bounded away from $f(x^\star)$ for all $t$. Note that under the conditions of the theorem, integrating the bound of Lemma 2, and using the asymptotic envelope of Lemma 3, gives

$$L(X(t), Z(t), t) - L(x_0, z_0, t_0) \leq -\int_{t_0}^t (f(X(\tau)) - f(x^\star))\eta(\tau)d\tau + \mathcal{O}(b(t)) + \mathcal{O}(\sqrt{b(t)\log\log b(t)}),$$

and if say $f(X(t)) - f(x^\star) \geq c > 0$ for all $t$, then the first term dominates the bound, and the energy would decrease to $-\infty$, a contradiction.

Combining these steps, we argue that $f(X(t))$ eventually becomes close to $f(x^\star)$ by (iii), then stays close by virtue of (i) and (ii). $\qquad\square$

The result of Theorem 1 makes it possible to guarantee almost sure convergence (albeit without guaranteeing a convergence rate) when the noise is persistent ($\sigma_*(t)$ is constant, or even increasing). To give a concrete example, suppose $\sigma_*(t) = \mathcal{O}(t^\alpha)$ (with $\alpha < \frac{1}{2}$ but can be positive), and let $\eta(t) = t^{-\alpha-\frac{1}{2}}$. Then $\eta(t)\sigma_*(t) = \mathcal{O}(t^{-\frac{1}{2}})$, $\int_{t_0}^t \eta(\tau)d\tau = \Omega(t^{-\alpha+\frac{1}{2}})$, $b(t) = \mathcal{O}(\log t)$, and $\sqrt{b(t)\log\log b(t)} = \mathcal{O}(\sqrt{\log t \log\log\log t})$, and the conditions of the theorem are satisfied. Therefore, with the appropriate choice of learning rate $\eta(t)$ (and the corresponding averaging in the primal space given by $a(t) = \eta(t)$), one can guarantee almost sure convergence.

Next, we derive explicit bounds on convergence rates. We start by bounding expected function values.

**Theorem 2.** *Suppose that $a = \eta/r$ and $\eta \geq \dot{r}$. Let $(X(t), Z(t))$ be the unique continuous solution to $\mathrm{SAMD}_{\eta,\eta/r,s}$. Then for all $t \geq t_0$,*

$$\mathbb{E}[f(X(t))] - f(x^\star) \leq \frac{L(x_0, z_0, t_0) + \psi(x^\star)(s(t) - s(t_0)) + \frac{nL_{\psi^*}}{2}\int_{t_0}^t \frac{\eta^2(\tau)\sigma_*^2(\tau)}{s(\tau)}d\tau}{r(t)}.$$

*Proof.* Integrating the bound of Lemma 2, and using the fact that $(f(X(t)) - f(x^\star))(\dot{r} - \eta) \leq 0$ by assumption on $\eta$, we have

$$L(X(t), Z(t), t) - L(x_0, z_0, t_0) \leq \psi(x^\star)(s(t) - s(t_0)) + \frac{nL_{\psi^*}}{2}\int_{t_0}^t \frac{\eta^2(\tau)\sigma_*^2(\tau)}{s(\tau)}d\tau + \int_{t_0}^t \langle V(\tau), dB(\tau)\rangle, \tag{13}$$

Taking expectations, the last term vanishes since it is an Itô martingale, and we conclude by observing that $\mathbb{E}[f(X(t))] - f(x^\star) \leq \mathbb{E}[L(X(t), Z(t), t)]/r(t)$. $\qquad\square$

To give a concrete example, suppose that $\sigma_*(t) = \mathcal{O}(t^\alpha)$ is given, and let $r(t) = t^\beta$ and $s(t) = t^\gamma$, $\beta, \gamma > 0$. To simplify, we will take $\eta(t) = \dot{r}(t) = \beta t^{\beta-1}$. Then the bound of Theorem 2 shows that $\mathbb{E}[f(X(t))] - f(x^\star) = \mathcal{O}(t^{\gamma-\beta} + t^{\beta+2\alpha-\gamma-1})$. To minimize the asymptotic rate, we can choose $\gamma - \beta = \beta + 2\alpha - \gamma - 1$, i.e. $\beta + \alpha - \gamma - \frac{1}{2} = 0$, and the resulting rate is $\mathcal{O}(t^{\alpha-\frac{1}{2}})$. In particular, we have:

**Corollary 2.** *Suppose that $\sigma_*(t) = \mathcal{O}(t^\alpha)$, $\alpha < \frac{1}{2}$. Then with $\eta(t) = (1-\alpha)t^{-\alpha}$, $a(t) = \frac{1-\alpha}{t}$, and $s(t) = t^{\frac{1}{2}}$, we have $\mathbb{E}[f(X(t))] - f(x^\star) = \mathcal{O}(t^{\alpha-\frac{1}{2}})$.*

**Remark 3.** *Corollary 2 can be interpreted as follows: Given a polynomial bound $\sigma_*(t) = \mathcal{O}(t^\alpha)$ on the volatility of the noise process, one can adapt the choice of primal and dual averaging rates ($a(t)$ and $\eta(t)$), which leads to an $\mathcal{O}(t^{\alpha-\frac{1}{2}})$ convergence rate.*

- *In the persistent noise regime ($\alpha = 0$), the dynamics use a constant $\eta$, and result in a $\mathcal{O}(1/\sqrt{t})$ rate. This rate is similar to the rectified dynamics proposed by Mertikopoulos and Staudigl [2016], but while they show convergence of the ergodic average $\tilde{X}(t) = \frac{1}{t}\int_0^t X(\tau)d\tau$, we can show convergence of the original process $X(t)$ under acceleration.*

- *In the vanishing noise regime ($\alpha < 0$), we can take advantage of the decreasing volatility by making $\eta(t)$ increasing. With the appropriate averaging rate $a(t)$, this leads to the improved rate $\mathcal{O}(t^{\alpha-\frac{1}{2}})$. It is worth observing here that when $\alpha \geq -\frac{1}{2}$, the same rate can be obtained for the ergodic average, without acceleration: one can show that the rectified SMD with $s(t) = t^{\max(0,\alpha+\frac{1}{2})}$ achieves a $\mathcal{O}(t^{\max(\alpha-\frac{1}{2},-1)})$. However for $\alpha < -\frac{1}{2}$, acceleration improves the rate from $\mathcal{O}(t^{-1})$ to $\mathcal{O}(t^{\alpha-\frac{1}{2}})$.*

- *In the increasing noise regime ($\alpha > 0$), as long as the volatility does not increase too fast ($\alpha < \frac{1}{2}$), one can still guarantee convergence by decreasing $\eta(t)$ with the appropriate rate.*

Finally, we give an estimate of the asymptotic convergence rate along solution trajectories.

**Theorem 3.** *Suppose that $a = \eta/r$ and $\eta \geq \dot{r}$. Let $(X(t), Z(t))$ be the unique continuous solution to* $\mathrm{SAMD}_{\eta, \eta/r, s}$. *Then*

$$
f(X(t)) - f(x^\star) = \mathcal{O}\left( \frac{s(t) + n \int_{t_0}^t \frac{\eta^2(\tau)\sigma_*^2(\tau)}{s(\tau)} + \sqrt{b(t) \log \log b(t)}}{r(t)} \right) \quad \text{a.s. as } t \to \infty,
$$

*where $b(t) = \int_{t_0}^t \eta^2(\tau)\sigma_*^2(\tau)d\tau$.*

*Proof.* Integrating the bound of Lemma 2 once again, we get inequality (13), where we can bound the Itô martingale term $\int_{t_0}^t \langle V(\tau), dB(\tau) \rangle$ using Lemma 3. This concludes the proof. □

Comparing the last bound to that of Theorem 2, we have the additional $\sqrt{b(t) \log \log b(t)}/r(t)$ term due to the envelope of the martingale term. This results in a slower a.s. convergence rate.

Suppose again that $\sigma_*(t) = \mathcal{O}(t^\alpha)$, and that $r(t) = t^\beta$ and $\eta(t) = \dot{r}(t) = \beta t^{\beta-1}$ to simplify. Then $b(t) = \int_{t_0}^t \eta^2(\tau)\sigma_*^2(\tau)d\tau = \mathcal{O}(t^{2\beta+2\alpha-1})$, and the martingale term becomes $\mathcal{O}(\sqrt{b(t) \log \log b(t)}/r(t)) = \mathcal{O}(t^{\alpha-\frac{1}{2}}\sqrt{\log \log t})$. Remarkably, the asymptotic rate of sample trajectories is, up to a $\sqrt{\log \log t}$ factor, the same as the asymptotic rate in expectation; one should observe, however, that the constant in the $\mathcal{O}$ notation is trajectory-dependent.

**Corollary 3.** *Suppose that $\sigma_*(t) = \mathcal{O}(t^\alpha)$, $\alpha < \frac{1}{2}$. Then with $\eta(t) = (1-\alpha)t^{-\alpha}$, $a(t) = \frac{1-\alpha}{t}$, and $s(t) = t^{\frac{1}{2}}$, we have $f(X(t)) - f(x^\star) = \mathcal{O}(t^{\alpha-\frac{1}{2}}\sqrt{\log \log t})$ a.s.*

## 5 Conclusion

Starting from the averaging formulation of accelerated mirror descent in continuous-time, and motivated by stochastic optimization, we formulated a stochastic variant and studied the resulting SDE. We discussed the role played by each parameter: the dual learning rate $\eta(t)$, the inverse sensitivity parameter $s(t)$, and the noise covariation bound $\sigma_*(t)$. Our results show that in the persistent noise regime, thanks to averaging, it is possible to guarantee a.s. convergence, remarkably even when $\sigma_*(t)$ is increasing (as long as $\sigma_*(t) = o(\sqrt{t})$). In the vanishing noise regime, adapting the choice of $\eta(t)$ to the rate of $\sigma_*(t)$ (with the appropriate averaging) leads to improved convergence rates, e.g. to $\mathcal{O}(t^{\alpha-\frac{1}{2}})$ in expectation and $\mathcal{O}(t^{\alpha-\frac{1}{2}}\sqrt{\log \log t})$ almost surely, when $\sigma_*(t) = \mathcal{O}(t^\alpha)$. These asymptotic bounds in continuous-time can provide guidelines in setting the different parameters of accelerated stochastic mirror descent.

It is also worth observing that in the deterministic case, one can theoretically obtain arbitrarily fast convergence, through a time change as observed by Wibisono et al. [2016] – a time-change would simply result in using different weights $\eta(t)$ and $a(t)$; this can also be seen in Corollary 1, where the rate $r(t)$ can be arbitrarily fast. In the stochastic dynamics, such a time-change would also lead to re-scaling the noise covariation, and does not lead to a faster rate. To some extent, adding the noise prevents us from "artificially" accelerating convergence using a simple time-change.

Finally, we believe this continuous-time analysis can be extended in several directions. For instance, it will be interesting to carry out a similar analysis for strongly convex functions, for which we expect faster convergence rates.

### Acknowledgments

We gratefully acknowledge the support of the NSF through grant IIS-1619362 and of the Australian Research Council through an Australian Laureate Fellowship (FL110100281) and through the Australian Research Council Centre of Excellence for Mathematical and Statistical Frontiers (ACEMS). We thank the anonymous reviewers for their insightful comments and suggestions.

## Footnotes

[1]In statistical learning, one seeks to minimize the expected risk (with respect to the true, unknown data distribution). A common approach is to minimize the empirical risk (observed on a given training set) then bound the distance between empirical and expected risk. Here we only focus on the optimization part.

[2]The initial conditions typically satisfy $\nabla\psi^*(z_0) = x_0$ which ensures that the trajectory starts with zero velocity, but this is not necessary in general.

[3]Such a $z^\star$ exists whenever $x^\star$ is in the relative interior of $\mathcal{X}$ (hence the condition $\mathcal{X}^\star \subset \text{relint } \mathcal{X}$ in Assumption 1). The analysis can be extended to minimizers that are on the relative boundary by replacing the Bregman divergence term in $L$ by the Fenchel coupling defined by Mertikopoulos and Staudigl [2016].

[4]In our model, we focus on the time dependence of the volatility. Note that in some settings, the variance of the gradient estimates scales with the squared norm of the gradient, see [Bottou et al., 2016] in the discrete case. Thus one can consider a model where $\sigma(x, t)$ scales with $\|\nabla f(x)\|_*$, which may lead to different rates.

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
