[Supplementary Material · acceleration_and_averaging_in_stochastic_descent_dynamics.pdf]

# Acceleration and Averaging In Stochastic Descent Dynamics Supplementary material, NIPS 2017

Walid Krichene        Peter Bartlett

## 1 Illustration of Stochastic Accelerated Mirror Descent

### 1.1 Construction of smooth mirror maps

The mirror map is central in defining mirror descent dynamics. We first give a generic method for constructing mirror maps, adapted to the feasible set $\mathcal{X} \subset E$. We fix a pair of dual reference norms, $\| \cdot \|, \| \cdot \|_*$, defined, respectively, on $E$ and its dual space $E^*$. We say that a map $F : E \to E^*$ is Lipschitz continuous on $\mathcal{X}$ with constant $L$ if for all $x, x' \in \mathcal{X}$, $\|F(x) - F(x')\|_* \leq L\|x - x'\|$. We recall that the effective domain of a convex function $\psi$ is the set $\{x \in E : \psi(x) < \infty\}$, and its convex conjugate $\psi^* : E^* \to \mathbb{R}$ is defined on $E^*$ by $\psi^*(z) = \sup_{x \in \mathcal{X}} \langle z, x \rangle - \psi(x)$. We recall that the sub-differential of $\psi$ at $x$ is the set $\partial\psi(x) = \{g \in E^* : \psi(x') \geq \psi(x) + \langle g, x' - x \rangle \ \forall x' \in \mathcal{X}\}$, and that $\psi$ is said to be $\mu$-strongly convex (w.r.t. $\| \cdot \|$) if $\forall x, x' \in \mathcal{X}$, $\forall g \in \partial\psi(x)$, $\psi(x) \geq \psi(x') + \langle g, x' - x \rangle + \frac{\mu}{2}\|x' - x\|^2$.

**Proposition 1.** *Let $\psi$ be a $\mu$-strongly convex function (w.r.t. $\|\cdot\|$) with effective domain $\mathcal{X}$, and let $\psi^*$ be its convex conjugate. Then $\psi^*$ is finite and differentiable on all of $E^*$, $\nabla\psi^*$ is $\frac{1}{\mu}$-Lipschitz, and has values in $\mathcal{X}$: specifically, for all $z \in E^*$,*

$$\nabla\psi^*(z) = \arg\max_{x \in \mathcal{X}} \langle z, x \rangle - \psi(x). \tag{1}$$

This follows from standard results from convex analysis, e.g. Theorems 13.3 and 25.3 in [Rockafellar, 1970]. To give an example of a Lipschitz mirror map, take $\psi$ to be the squared Euclidean norm, $\psi(x) = \frac{1}{2}\|x\|_2^2$. Then $\psi^*(z) = \arg\max_{x \in \mathcal{X}} \langle z, x \rangle - \frac{1}{2}\|x\|_2^2 = \arg\min_{x \in \mathcal{X}} \|z - x\|_2^2$, and the mirror map reduces to the Euclidean projection on $\mathcal{X}$. It is worth noting that although one can theoretically construct a smooth mirror map given any convex feasible set $\mathcal{X}$, using Proposition 1, this does not necessarily mean that the mirror map can be implemented efficiently, since in its general form, it is given by the solution to the problem (1). However, many convex sets have known mirror maps that are efficient to compute. For a concrete example, when $\mathcal{X}$ is the probability simplex $\Delta = \{x \in \mathbb{R}_+^n : \sum_{i=1}^n x_i = 1\}$, choosing $\psi$ to be the negative entropy $\psi(x) = \sum_{i=1}^n x_i \log x_i$ yields a closed-form mirror map given by $(\nabla\psi^*(z))_i = e^{z_i} / \sum_{j=1}^n e^{z_j}$, see e.g. Banerjee et al. [2005] for additional examples.

### 1.2 Stochastic Accelerated Mirror Descent

$$\text{SAMD}_{\eta,a,s} \begin{cases} dZ(t) = -\eta(t)[\nabla f(X(t))dt + \sigma(X(t),t)dB(t)] \\ dX(t) = a(t)[\nabla\psi^*(Z(t)/s(t)) - X(t)]dt, \end{cases} \tag{2}$$

We give an illustration of SAMD dynamics in Figure 1, on a simplex-constrained problem in $\mathbb{R}^3$. The feasible set is given by $\mathcal{X} = \{x \in \mathbb{R}_+^n : \sum_{i=1}^n x_i = 1\}$, and the mirror map is generated by the negative entropy restricted to the simplex, given by:

$$\psi(x) = \begin{cases} -\sum_{i=1}^n x_i \ln x_i & \text{if } x \in \mathcal{X}, \\ +\infty & \text{otherwise,} \end{cases}$$

which is strongly convex w.r.t. the norm $\|\cdot\|_1$ by Pinsker's inequality. Its convex conjugate is given by

$$\psi^*(z) = \max_{x \in \mathcal{X}} \langle x, z \rangle - \sum_{i=1}^{n} x_i \ln x_i = \ln \sum_{i=1}^{n} e^{z_i}$$

which is differentiable for all $z \in E^*$. The mirror map satisfies

$$\nabla\psi^*(z)_i = \frac{e^{z_i}}{\sum_{j=1}^{n} e^{z_j}}$$

and so is Lipschitz w.r.t. the dual norms $\|\cdot\|_1, \|\cdot\|_\infty$.

Note that for all $z \in E^*$ and all $\alpha \in \mathbb{R}$,

$$\nabla\psi^*(z) = \nabla\psi^*(z + \alpha\mathbf{1}),$$

where $\mathbf{1}$ is the vector of all ones. This can also be seen as a consequence of the duality of sub-differentials (e.g. Theorem 23.5 in Rockafellar [1970]), which states that $x = \nabla\psi^*(z)$ if and only if $z \in \partial\psi(x)$, and since $\psi$ is the restriction of the negative entropy $-H(x) = -\sum_{i=1}^{n} x_i \ln x_i$ to the simplex, its sub-differential at $x$ is

$$\partial\psi(x) = -\nabla H(x) + n_{\mathcal{X}}(x)$$

where $n_{\mathcal{X}}(x)$ is the normal cone to $\mathcal{X}$ at $x$, which is simply the line $\mathbb{R}\mathbf{1}$ (when $x$ is in the relative interior of the simplex).

Since the mirror map is constant along the normal to the simplex, we choose to project the dual variable $Z$ on the hyperplane parallel to the simplex, for visualization purposes. This allows us to visualize the relevant component of the dual dynamics, and ignore a component which does not matter for convergence (but which could have high magnitudes if $\nabla f$ has a large component along the normal). Note that even numerically, projecting $Z$ after each iteration helps improve numerical stability (without affecting the primal trajectory).

Finally in order to visualize the function values, we generate a triangular mesh of the simplex, then map it to the dual space. In other words, the colors in the primal space represent $f(x)$, and in the dual space represent $f(\nabla\psi^*(z))$. It is interesting to observe how the mirror map $\nabla\psi^*$ distorts the space between primal and dual spaces.

Figure 1: Illustration of SAMD dynamics. The dual trajectory $Z(t)$ cumulates negative gradients, $\dot{Z}(t) = -\eta(t)\nabla f(X(t))$. We visualize the scaled dual trajectory $Z(t)/s(t)$ in the dual space (red), and the corresponding mirror $\nabla\psi^*(\frac{Z(t)}{s(t)})$ in the primal space (dotted red). The primal trajectory $X(t)$ is obtained by averaging the mirror.

# 2 Proof of Theorem 1

The proof of Theorem 1 follows a similar outline to that of Theorem 4.1 in [Mertikopoulos and Staudigl, 2016], with some significant changes: first, we do not make the assumption that the minimizer is unique. Second, and most importantly, the dynamics and the energy function are different; the dual learning rate $\eta(t)$ and the primal averaging $a(t)$ are essential in our case to handle the noise, since we do not assume that the volatility bound $\sigma_*(t)$ is vanishing (i.e. we do not operate under a small noise limit assumption). This introduces important changes to the argument.

First, we recall the definition of an asymptotic pseudo trajectory (APT), due to Benaïm [1999], and adapt it to our setting.

**Definition** (Asymptotic Pseudo Trajectory). *Let $\Phi_t : \mathcal{X} \times E^* \to \mathcal{X} \times E^*$ be the semi-flow associated with the deterministic dynamics $\mathrm{AMD}_{\eta,\eta,1}$, that is, $(x(t), z(t)) = \Phi_t(x_0, z_0)$ is the solution of the deterministic dynamics $\mathrm{AMD}_{\eta,\eta,1}$ with initial condition $(x_0, z_0)$. A continuous function $t \mapsto (X(t), Z(t)) \in \mathcal{X} \times E^*$ is an asymptotic pseudo trajectory (APT) for $\Phi_t$ if for all $T > 0$,*

$$\lim_{t \to \infty} \sup_{0 \le h \le T} d((X(t+h), Z(t+h)), \Phi_h(X(t), Z(t))) = 0,$$

*where $d$ is a distance on $\mathcal{X} \times E^*$, e.g. $d((x, z), (x', z')) = \|x - x'\| + \|z - z'\|_*$.*

Next, we specialize the energy function, and the bounds on its time derivative, both for the deterministic and stochastic dynamics. Under the assumptions of the theorem $(r(t) = s(t) = 1)$, $L(x, z, t)$ simplifies to

$$L_{z^\star}(x, z) = f(x) - f(x^\star) + D_{\psi^*}(z, z^\star),$$

where we added the subscript $z_\star$ to insist on the fact that the energy function is "anchored" at $z^\star$. Note that since the minimizer is not necessarily unique, $L_{z^\star}(x(t), z(t))$ does not necessarily converge to 0 for arbitrary $z^\star$. Thus, we define and use

$$\bar{L}(x, z) = \inf_{z^\star \in \mathcal{Z}^\star} L_{z^\star}(x, z),$$

where $\mathcal{Z}^\star = \{z^\star \in E^* : \nabla\psi^*(z^\star) \in \mathcal{X}^\star\} = \cup_{x^\star \in \mathcal{X}^\star} \partial\psi(x^\star)$ (by the fact that $x^\star \in \partial\psi^*(z^\star)$ if and only if $z^\star \in \partial\psi(x^\star)$).

Next, we observe that since $\nabla f$ is $L_f$-Lipschitz and $\nabla\psi^*$ is $L_{\psi^*}$-Lipschitz, we can bound the change of the energy due to small displacements in $(x, z)$: we will use the fact that for any convex function $f$ with $L$-Lipschitz gradient, $f(x + \delta_x) \le f(x) + \langle \nabla f(x), \delta_x \rangle + \frac{L}{2}\|\delta_x\|^2$. We have

$$L_{z^\star}(x + \delta_x, z + \delta_z) = f(x + \delta_x) - f(x^\star) + \psi^*(z + \delta_z) - \psi^*(z^\star) - \langle \nabla\psi^*(z^\star), z + \delta_z - z^\star \rangle$$

$$\le f(x) + \langle \nabla f(x), \delta_x \rangle + \frac{L_f}{2}\|\delta_x\|^2 - f(x^\star)$$

$$+ \psi^*(z) + \langle \nabla\psi^*(z), \delta_z \rangle + \frac{L_{\psi^*}}{2}\|\delta_z\|_*^2 - \psi^*(z^\star) - \langle \nabla\psi^*(z^\star), z + \delta_z - z^\star \rangle$$

$$= L_{z^\star}(x, z) + \langle \nabla f(x), \delta_x \rangle$$

$$+ \frac{L_f}{2}\|\delta_x\|^2 + \langle \nabla\psi^*(z) - \nabla\psi^*(z^\star), \delta_z \rangle + \frac{L_{\psi^*}}{2}\|\delta_z\|_*^2$$

$$\le L_{z^\star}(x, z) + G\|\delta_x\| + \frac{L_f}{2}\|\delta_x\|^2 + D\|\delta_z\|_* + \frac{L_{\psi^*}}{2}\|\delta_z\|_*^2 \qquad (3)$$

where in the last inequality, $G = \sup_{x \in \mathcal{X}} \|\nabla f(x)\|_*$ (which is bounded since $\nabla f$ is continuous and $\mathcal{X}$ is compact), and $D$ is the diameter of $\mathcal{X}$.

For the deterministic dynamics, the bound of Lemma 1 becomes

$$\frac{d}{dt} L_{z^\star}(x(t), z(t)) \le -\eta(t)(f(x(t)) - f(x^\star)), \qquad (4)$$

and for the stochastic dynamics, the bound of Lemma 2 becomes

$$dL_{z^\star}(X(t), Z(t)) \le \left[ -\eta(t)(f(X(t)) - f(x^\star)) + \frac{L_{\psi^*}}{2}\eta^2(t)\sigma_*^2(t) \right] dt + \langle V(t), dB(t) \rangle. \qquad (5)$$

We now proceed according to the steps outlined in the proof sketch. We give an illustration of the argument in Figure 2.

(i) We start by proving that under the conditions of Theorem 1, the stochastic process $(X(t), Z(t))$ (that is, the unique continuous solution of the stochastic dynamics $\text{SAMD}_{\eta,\eta,1}$) is an APT for the deterministic semi-flow of $\text{AMD}_{\eta,\eta,1}$. Since the volatility term is $-\eta(t)\sigma(X(t), t)dB(t)$, it suffices, by Proposition 4.6[1] in [Benaïm, 1999], to show that $\int_{t_0}^{\infty} e^{-\frac{c}{\eta^2(t)\sigma_*^2(t)}}$ is finite for all $c > 0$. But we have, by assumption, $\eta(t)\sigma_*(t) = o(1/\sqrt{\log t})$, thus $\eta^2(t)\sigma_*^2(t) = \epsilon(t)/\log t$ with $\lim_{t\to\infty} \epsilon(t) = 0$, and $\int_{t_0}^{\infty} e^{-\frac{c}{\eta^2(t)\sigma_*^2(t)}} dt = \int_{t_0}^{\infty} e^{-\frac{c\log t}{\epsilon(t)}} dt = \int_{t_0}^{\infty} t^{-\frac{c}{\epsilon(t)}} dt$, which is finite.

We also show that by virtue of the APT property (and the fact that the energy function is Lipschitz), we can bound the difference between the energy $\bar{L}$ along deterministic and stochastic solutions starting at the same point. Indeed, Inequality (3) shows that $L_{z^\star}(x + \delta_x, z + \delta_z) - L_{z^\star}(x, z) \leq \epsilon$ whenever $\max(\|\delta_x\|, \|\delta_z\|_*)$ is small enough. Therefore, by the APT property, for all $\epsilon > 0$ and all $T > 0$, there exists $t_T$ such that for all $t \geq t_T$ and all $h \in [0, T]$,

$$L_{z^\star}(X(t+h), Z(t+h)) - L_{z^\star}(\Phi_h(X(t), Z(t))) \leq \epsilon/2,$$

and this holds uniformly over $z^\star$. In particular, since $\bar{L}$ is defined to be the infimum over all $z^\star$, we can find some $z_0^\star$ such that

$$L_{z_0^\star}(\Phi_h(X(t), Z(t))) \leq \bar{L}(\Phi_h(X(t), Z(t))) + \epsilon/2.$$

Then

$$\begin{aligned}
\bar{L}(X(t+h), Z(t+h)) &\leq L_{z_0^\star}(X(t+h), Z(t+h)) \\
&\leq L_{z_0^\star}(\Phi_h(X(t), Z(t))) + \epsilon/2 \\
&\leq \bar{L}(\Phi_h(X(t), Z(t))) + \epsilon.
\end{aligned}$$

(ii) Next, we prove a stability property of the energy for the deterministic dynamics. Fix $\epsilon > 0$ and let $V_\epsilon = \{(x, z) : \bar{L}(x, z) \leq \epsilon\}$. Then $\Phi_t(x, z) \in V_\epsilon$ if $(x, z) \in V_\epsilon$ (since $\bar{L}$ is non-increasing, as the infimum of non-increasing functions). Besides, we claim that there exists $T > 0$ such that for all $t \geq T$,

$$\bar{L}(\Phi_t(x, z)) \leq \min(\epsilon, \bar{L}(x, z) - \epsilon).$$

Indeed, by continuity of $f$, there exists $c > 0$ such that $f(x) - f(x^\star) > c$ for all $(x, z) \notin V_\epsilon$, and integrating the bound (4), we have, for all $z^\star$,

$$L_{z^\star}(\Phi_t(x, z)) \leq L_{z^\star}(x, z) - c\int_T^t \eta(\tau)d\tau,$$

therefore, setting $T = 2\epsilon/c$, we know that either $\Phi_t(x, z) \in V_\epsilon$ for some $t_1 \leq T$, in which case the trajectory remains in $V_\epsilon$ after $t_1$, or $\Phi_t(x, z)$ remains outside of $V_\epsilon$, in which case $L_{z^\star}(\Phi_T(x, z)) \leq L_{z^\star}(x, z) - 2\epsilon$ for all $z^\star$. Since $\bar{L}$ is defined to be the infimum over all $z^\star$, we can find some $z_0^\star$ such that $L_{z_0^\star}(x, z) \leq \bar{L}(x, z) + \epsilon$. Then

$$\bar{L}(\Phi_T(x, z)) \leq L_{z_0^\star}(\Phi_T(x, z)) \leq L_{z_0^\star}(x, z) - 2\epsilon \leq \bar{L}(x, z) - \epsilon.$$

(iii) Next, we prove that the stochastic process cannot stay outside of $V_\epsilon$ for unbounded intervals of time. Indeed, fix $\epsilon > 0$, and $T > 0$, and suppose that with positive probability, $(X(t), Z(t))$ remains outside $V_\epsilon$ for all $t \geq T$. Then by continuity of $f$, there exists $c > 0$ such that $f(X(t)) - f(x^\star) \geq c$ for all $t \geq T$, and integrating the bound (5) gives

$$\begin{aligned}
&L_{z^\star}(X(t), Z(t)) - L_{z^\star}(X(T), Z(T)) \\
&\leq -c\int_T^t \eta(\tau)d\tau + \mathcal{O}(b(t)) + \mathcal{O}(\sqrt{b(t)\log\log b(t)}),
\end{aligned}$$

where the right-hand side converges to $-\infty$ since, by assumption, $\int_{t_0}^{t} \eta(\tau)d\tau$ dominates $b(t)$ and $\sqrt{b(t)\log\log b(t)}$. This would imply that, with positive probability, $L(X(t), Z(t), t) \to -\infty$, a contradiction. Therefore, for all $\epsilon > 0$ and for all $T$, there exists $t \geq T$ such that $(X(T), Z(T)) \in V_\epsilon$ a.s.

We are now ready to put together the different parts of the argument. Fix $\epsilon > 0$. By (ii), there exists $T_0$ such that

$$\bar{L}(\Phi_{T_0}(x, z)) \leq \min(\epsilon/3, \bar{L}(x, z) - \epsilon/3). \tag{6}$$

By (i) there exists $T_1$ such that for $t \geq T_1$ and for all $h \in [0, T_0]$,

$$\bar{L}(X(t+h), Z(t+h)) \leq \bar{L}(\Phi_h(X(t), Z(t))) + \epsilon/3, \tag{7}$$

By (iii), there exists $T_2 \geq \max(T_0, T_1)$ such that $(X(T_2), Z(T_2)) \in V_{\epsilon/3}$.

Now we show that the trajectory remains in $V_\epsilon$ for all $t \geq T_2$. Indeed, by induction on $k$, we have $\bar{L}(X(T_2 + kT_0), Z(T_2 + kT_0)) \leq 2\epsilon/3$ for all $k \in \mathbb{N}$ (by (6) and (7)). Then for all $h \in [0, T_0]$,

$$\bar{L}(X(T_2 + kT_0 + h), Z(T_2 + kT_0 + h)) \leq \bar{L}(\Phi_h(X(T_2 + kT_0), Z(T_2 + kT_0))) + \epsilon/3$$
$$\leq 2\epsilon/3 + \epsilon/3.$$

Since $\epsilon$ is arbitrary, this proves that for all $\epsilon$ $(X(t), Z(t))$ remains in $V_\epsilon$ for $t$ large enough, a.s. But by definition of $\bar{L}$, $(x, z) \in V_\epsilon$ implies that $f(x) - f(x^\star) \leq \epsilon$, which proves that $f(X(t)) - f(x^\star)$ converges to 0 a.s. $\qquad\square$

Figure 2: Illustration of the proof of Theorem 1, with $\epsilon = 2.4 \times 10^{-3}$, and $T_0 = 20$. The right plot shows in blue the value of the energy function $\bar{L}(X(t), Z(t))$ along one sample trajectory $(X(t), Z(t))$ of the SAMD dynamics; and in green the energy function along solutions of the deterministic ODE $\{(x_k(t), z_k(t)), t \in [T_2 + kT_0, T_2 + (k+1)T_0]\}$, initialized at $(X(T_2 + kT_0), Z(T_2 + kT_0))$. We also highlight a cylinder of radius $\frac{\epsilon}{3}$ centered at the deterministic energy. Note that for large enough times, the sample path of the stochastic dynamics remains within the cylinder. The dashed lines show the energy levels $\frac{\epsilon}{3}$, $\frac{2\epsilon}{3}$, and $\epsilon$. Finally, the left plot visualizes these trajectories in the primal space (where we used a different color for each interval $[T_2 + kT_0, T_2 + (k+1)T_0]$).

# 3 Dynamics of Nesterov's accelerated method

Nesterov's accelerated method [Nesterov, 1983] has been shown by Su et al. [2014] to be the discretization of the ODE:

$$\ddot{x}(t) = -\nabla f(x(t)) - \frac{\alpha}{t}\dot{x}(t), \tag{8}$$

with $\alpha \geq 3$, which describes the motion of a damped non-linear oscillator, driven by the potential $f$, and subject to a viscous friction term $\frac{\alpha}{t}\dot{X}$. Cabot et al. [2009] had

previously studied a general family of such damped oscillators with vanishing friction, but the connection with Nesterov's method was not made until [Su et al., 2014]. Note that the dynamics are unconstrained in this case. This ODE can be recovered as a special case of AMD, for which we restate the general form below (and assume $s \equiv 1$ for simplicity):

$$\text{AMD} \begin{cases} \dot{z}(t) = -\eta(t)\nabla f(x(t)) \\ \dot{x}(t) = a(t)(\nabla\psi^*(z(t)) - x(t)). \end{cases} \tag{9}$$

First, we take the mirror map to be the identity, which corresponds to taking $\psi(x) = \frac{1}{2}\|x\|_2^2$. Then, writing the second equation of AMD as $\frac{\dot{x}(t)}{a(t)} = z(t) - x(t)$ and taking derivatives, we have

$$\frac{1}{a(t)}\ddot{x}(t) - \frac{\dot{a}(t)}{a^2(t)}\dot{x}(t) = \dot{z}(t) - \dot{x}(t) = -\eta(t)\nabla f(x(t)) - \dot{x}(t),$$

i.e.

$$\ddot{x}(t) = -\eta(t)a(t)\nabla f(x(t)) - \dot{x}(t)\frac{a^2(t) - \dot{a}(t)}{a(t)},$$

and by taking $r(t) = \frac{t^2}{\beta^2}$, $\eta(t) = \frac{t}{\beta}$, $a(t) = \frac{\beta}{t}$, with $\beta \geq 2$, the ODE becomes

$$\ddot{x}(t) = -\nabla f(x(t)) - \dot{x}(t)\frac{\beta+1}{t},$$

which is equivalent to Nesterov's ODE (8) up to the reparameterization $\alpha = \beta + 1$. It is easy to verify the conditions $\eta \geq \dot{r}$ and $a = \eta/r$ when $\beta \geq 2$, thus as a consequence of Corollary 1, $f(x(t)) - f(x^\star) = \mathcal{O}(1/r(t)) = \mathcal{O}(1/t^2)$, which is analogous to the quadratic rate of Nesterov's accelerated method in discrete time.

# 4   Effects of time-change

In the deterministic case, one can obtain arbitrarily fast convergence through a time change, as observed by Wibisono et al. [2016] – a time-change would simply result in using different weights $\eta(t)$ and $a(t)$. In the stochastic dynamics, such a time-change would also lead to re-scaling the noise co-variation, and does not lead to a faster rate. To some extent, adding the noise prevents us from "artificially" accelerating convergence using a simple time-change. To illustrate this difference, first consider a time-change in the deterministic case. Let $(x(t), z(t))$ be the unique solution to $\text{AMD}_{\eta,a,1}$, which we rewrite below (where we took $s(t) \equiv 1$ to simplify),

$$\begin{cases} \dot{z}(t) = -\eta(t)\nabla f(x(t)) \\ \dot{x}(t) = a(t)(\nabla\psi^*(z(t)) - x(t)), \end{cases}$$

and consider a differentiable increasing function of time, $\gamma(t)$. Let $(x', z')$ be defined by $x'(t) = x(\gamma(t))$ and $z'(t) = z(\gamma(t))$. Then $(x', z')$ satisfy the following dynamics:

$$\begin{cases} \dot{z}'(t) = -\dot{\gamma}(t)\eta(\gamma(t))\nabla f(x'(t)) \\ \dot{x}'(t) = \dot{\gamma}(t)a(\gamma(t))[\nabla\psi^*(z'(t)) - x'(t)], \end{cases}$$

thus $(x', z')$ is the unique solution to $\text{AMD}_{\tilde{\eta},\tilde{a},1}$ where $\tilde{\eta}(t) = \dot{\gamma}(t)\eta(\gamma(t))$ and $\tilde{a}(t) = \dot{\gamma}(t)a(\gamma(t))$, and if $\gamma$ is super linear, $f(x'(t))$ will have a faster convergence rate than $f(x(t))$. Indeed, if $\eta, a, r$ satisfy the conditions of Corollary 1 (i.e. $a = \eta/r$ and $\eta \geq \dot{r}$), then $f(x(t)) - f(x^\star) \leq \frac{L(x_0,z_0,t_0)}{r(t)}$; but $\tilde{\eta} = \dot{\gamma}\eta \circ \gamma, \tilde{a} = \dot{\gamma}a \circ \gamma, \tilde{r} = r \circ \gamma$ also satisfy the conditions of the corollary, thus

$$f(x'(t)) - f(x^\star) \leq \frac{L(x_0, z_0, t_0)}{r(\gamma(t))}.$$

Let us now consider a similar time-change in the stochastic case. Let $(X, Z)$ be the unique (a.s.) continuous solution of $\text{SAMD}_{\eta, \eta/r, s}$, which we rewrite below.

$$\text{SAMD}_{\eta, a, 1} \begin{cases} dZ(t) = -\eta(t)dG(t) \\ dX(t) = a(t)[\nabla\psi^*(Z(t)) - X(t)]dt, \end{cases} \quad (10)$$

where $G(t)$ is the noisy gradient process, defined by

$$dG(t) = \nabla f(X(t))dt + \sigma(X(t), t)dB(t),$$

and has covariation

$$d[G_i(t), G_j(t)] = (\sigma(X(t), t)\sigma(X(t), t)^T)_{i,j}dt = \Sigma_{ij}(X(t), t)dt. \quad (11)$$

Define $(X', Z')$ by the (differentiable, increasing) time-change $X'(t) = X(\gamma(t))$ and $Z'(t) = Z(\gamma(t))$. Then, using the following time-change identity for Itô martingales (see, e.g. Lemma 2.3 in [Kobayashi, 2011]):

$$\int_{\gamma(t_0)}^{\gamma(t)} \sigma(X(\tau), \tau)dB(\tau) = \int_{t_0}^{t} \sigma(X(\gamma(\tau)), \gamma(\tau))dB(\gamma(\tau)),$$

we have

$$\begin{cases} dZ'(t) = -\eta(\gamma(t))[\nabla f(X'(t))\dot{\gamma}(t)dt + \sigma(X'(t), \gamma(t))dB(\gamma(t))] \\ dX'(t) = a(\gamma(t))[\nabla\psi^*(Z'(t)) - X'(t)]\dot{\gamma}(t)dt, \end{cases}$$

which we can rewrite as

$$\begin{cases} dZ'(t) = -\tilde{\eta}(t)d\tilde{G}(t) \\ dX'(t) = \tilde{a}(t)[\nabla\psi^*(Z'(t)) - X'(t)]dt. \end{cases}$$

where $\tilde{\eta}, \tilde{a}$ are as defined in the deterministic case, and $\tilde{G}$ is defined by

$$d\tilde{G}(t) = \nabla f(X'(t)) + \frac{\sigma(X'(t), \gamma(t))}{\dot{\gamma}(t)}dB(\gamma(t)).$$

In particular, we observe that the noise covariation of $\tilde{G}$ is

$$d[\tilde{G}_i(t), \tilde{G}_j(t)] = \frac{1}{\dot{\gamma}(t)^2}\Sigma_{ij}(X'(t), \gamma(t))\dot{\gamma}(t)dt \quad (12)$$

where we used the fact that the time-changed Brownian motion $B(\gamma(t))$ has quadratic covariation $d[B_i(\gamma(t)), B_i(\gamma(t))] = \dot{\gamma}(t)dt$.

Comparing the quadratic covariation of $G$ and $\tilde{G}$ (equations (11) and (12) respectively), it becomes apparent that, unless $\gamma$ is the identity, rescaling time also rescales the covariation of the noise (even in the case where $\Sigma(x, t)$ does not depend on $t$, due to the $\dot{\gamma}(t)$ term). In other words, accelerating time by $\gamma(t)$ would scale down the variance of the gradient by $\dot{\gamma}(t)$, and $(X', Z')$ would not be a solution to the original problem anymore, unlike in the deterministic case.

## Footnotes

[1] Proposition 4.6 in [Benaïm, 1999] is stated in terms of solutions to a martingale problem, which is equivalent to solutions to the SDE; see, for example, [Stroock and Varadhan, 1972].