[Reviews · NeurIPS 2017]

Reviewer 1



Summary: The authors consider mirror descent dynamics in continuous time for constrained optimization problems. They work in the context of stochastic differential equations and consider parametrized dynamics which include many of the recently considered parameters for such dynamics. This includes time dependant learning rate, sensitivity and averaging parameters which provide a great flexibility in the analysis. The dynamics also feature acceleration terms which were recently proposed in a deterministic setting. In a first step, the authors carry out an asymptotic analysis in the deterministic setting. They then introduce a stochastic variant of the dynamics and describe uniqueness properties of its solutions. Combining properties of the deterministic dynamics and stochastic calculus the authors propose three results: Almost sure convergence of the sample trajectories, convergence rates, both in expectation and for sample trajectories. Main comments: First of all, I have a limited knowledge of stochastic differential calculus and do not feel competent to validate all the technical details in the manuscript, although none of them felt particularly awkward. I believe that this work is of very good quality. Technical elements are clearly presented and the authors do not neglect fundamental aspects of the problem such as existence and uniqueness. Furthermore, the overall exposition is clear and of good quality and the text is well balanced between the main proof arguments and further details given in the appendix. To my understanding, the convergence analysis carried out by the authors extends significantly existing results from the litterature. The model is flexible enough to allow many possible choice of the dynamics. For example the authors are able to show that unit learning rate may robustly maintain good properties of accelerated dynamics in the deterministic case provided that suitable averaging is performed. Overall, I think that this constitutes valuble work and should be considered for publication. Additional comments and questions: 1- The authors motivate the introduction of stochastic dynamics by emphasizing on stochastic methods for finite sums. However the additive noise structure is not particularly well suited to model such cases. Indeed, in these settings the variance of the noise usually features a multiplicative term. 2- All the results are presented in abstract form with the addition of an other parameter "r" and assumptions on it. It is not really clear how the assumptions on r restrict the choice of the other parameters. As a result, the convergence rates are not easily read or interpreted. Beyon the proposed corollaries, it is not really clear how the given result provide interesting guaranties in terms of function values. For example would the work of the authors allow to get better rates in the small noise regime? 3- Do the authors have any comment on how the analysis which is carried out in a stochastic, continuous time setting could impact the discrete time setting. Is there a clear path to the conception of efficient algorithms and in particular is it possible to provide a discrete time equivalent for the different time dependent parameters in the ODE?

Reviewer 2



The paper formulates a stochastic differential equation framework to study the accelerated mirror descent dynamics in the continuous time settings. The main contribution of the paper is to provide convergence analysis of such dynamical system under noisy gradient, where a random Brownian process is presented in the gradient evaluation. The presentation of the paper is very clear and well organized and the theoretical result is solid. The only terminology not clear to me is the term averaging formulation, can authors provide more explanations on it? The convergence result is very interesting because it provides intuition about how the error's magnitude influence the choice of parameters. One question I would like to ask is how can we discretize the given continuous dynamics and what will be the representations of different parameters in the discretized setting. Moreover, since the provided stochastic dynamics is about the accelerated mirror descent, I am wondering is there any acceleration in terms of convergence rate is obtained compare to the standard mirror descent dynamics, like the one in Mertikopoulos and Staudigl 2016. Can authors comment on it? Overall, I find the paper very clear and it is theoretically solid. I am happy to accept it if authors can address my concerns. Remarks: There is a tiny impreciseness in the equation after line 217 in page 6, it should be a limsup instead of limit, it doesn't affect the correctness of the proof. Some other typos appears in the equation after line 147 in page 4. #EDIT Clarified by the authors. #EDIT Thank you for author's feedback, I believe this is a good paper to be published, I vote for accept.

Reviewer 3



This paper studies stochastic dynamics for optimizing smooth convex functions beyond the Euclidean framework. The authors use technique introduced by Raginsky (2012) to study a stochastic modification of the accelerated mirror descent studied by Krichene (2015). The paper is very clear and well written. It is extremely pleasant to read and the related work is seriously done. To conclude, this paper takes up the challenge to understand stochastic optimization algorithms thanks to its comparison with its continuous counterpart which satisfies a SDE. This follows recent progress in the understanding of Accelerated gradient descent (see e,g,. Su, 2015). Even if stochastic calculus is an involved field of mathematics, this paper succeeds in proving interesting result. Comments: - Why put so much effort to treat the mirror descent case whereas the Euclidean case is not already done ? - What does acceleration bring to the story: a clear comparison of the results with the non-accelerated ones would be enlightening. - The propositions are not explained and commented enough. - The rates and the conditions could be further detailed - 155: The corollary 1 deserves further comments: What does the condition $\eta \geq \dot r$ mean ? Which rates give this result ? What is the dependency on $L$ ? Why this is an accelerated result ? - 183: Proposition 2: What are the conditions on the algorithm parameters to have convergence ? Minor comments: 14: E=\RR^d, E is not necessary, just \RR^d 50: This part is misleading. The true problem of statistical learning is directly to optimize the true expectation with respect to the unknown law of the data. And this is directly what optimize stochastic gradient descent. The finite sum problem is a different and simpler problem (with different same lower bounds). 50-60: What about analysis of regular gradient descent in continuous time ? 68: Could the authors give some examples where the noise is decreasing like this. Sec 2.1 This section is useless. 128-130: What are the links between s(t) and \eta(t) ? 137: 1992 is not the correct citation date. Maybe 1892. 1992 corresponds to some reissue. 146: Given that $a$ and $\eta$ are fixed by the algorithm and that $r(t)$ parametrized the Lyapunov function, I would rather write $r=\eta/a$.